# Live Yeast (*Saccharomyces cerevisiae* var. *boulardii*) Supplementation in a European Sea Bass (*Dicentrarchus labrax*) Diet: Effects on the Growth and Immune Response Parameters

**DOI:** 10.3390/ani13213383

**Published:** 2023-10-31

**Authors:** Anna Perdichizzi, Martina Meola, Letteria Caccamo, Gabriella Caruso, Francesco Gai, Giulia Maricchiolo

**Affiliations:** 1Institute for Marine Biological Resources and Biotechnology, National Research Council (CNR-IRBIM), Spianata S. Raineri, 98122 Messina, Italy; anna.perdichizzi@cnr.it (A.P.); letteria.caccamo@cnr.it (L.C.); giulia.maricchiolo@cnr.it (G.M.); 2Institute of Polar Sciences (CNR), Spianata S. Raineri, 98122 Messina, Italy; gabriella.caruso@cnr.it; 3Institute of Sciences of Food Production (CNR), Largo Paolo Braccini, 10095 Grugliasco, Italy; francesco.gai@ispa.cnr.it

**Keywords:** probiotics, gut immunity, antibacterial activity, *Dicentrarchus labrax*

## Abstract

**Simple Summary:**

In animal science, dietary supplementation with probiotics offers several advantages, in terms of both environmental sustainability and production, and probiotic supplementation in aquatic animals is known to enhance stress tolerance, disease prevention, and growth, as has already been documented for several fish species. Among the various probiotics, the yeast-based ones have been stimulating a great deal of interest because of their ability to produce polyamines, which participate in numerous biological processes, and *Saccharomyces cerevisiae* var. *boulardii* (LSB), one of the most studied yeasts, has been used throughout the world for the prevention and treatment of a variety of gut inflammatory diseases. In our study, in order to obtain insights into the potential beneficial effects of this yeast as a probiotic, the dietary inclusion of an LSB strain (CNCM I-1079) was evaluated, at increasing concentrations (0, 100, and 300 mg kg^−1^ of feed) for 90 days, on the welfare and health status of juvenile European sea bass, an important fish species in Mediterranean aquaculture. Our results indicate that LSB has an immunomodulatory action, and the observed lowering of pro-inflammatory gene transcripts in the gut suggests a possible anti-inflammatory action of this probiotic strain on the gut immune system of sea bass, with a possible enhancement of fish disease prevention. LSB also exerted a low, albeit insignificant, stimulating effect on the hematological and immunological parameters.

**Abstract:**

The present study has been aimed at evaluating the effects of the dietary inclusion of the live yeasts, *Saccharomyces cerevisiae* var. *boulardii* (LSB) administered at increasing concentrations (0, 100, and 300 mg kg^−1^ of feed, here referred to as LSB 0, 100, 300) for 90 days, on the health conditions of European sea bass. The main zootechnical parameters, histological and morphological analyses, innate immunity response parameters (intestinal cytokine expression, lysozyme content, spontaneous hemolytic and hemagglutinating activities, antibacterial activities, and peroxidase activity) were measured as fish welfare parameters. LSB did not impair either growth parameters or the morphometric indexes. LSB down-regulated interleukin-1β transcription in the distal gut of fish treated with 5.4 × 10^5^ CFU g^−1^ (LSB100) for 21 days. The interleukin-6 mRNA level decreased significantly in the proximal gut for both doses of yeast, after 21 days of feeding; the gene expression of interleukin-6 was significantly lower in the sea bass fed 10.81 × 10^5^ CFU g^−1^ (LSB300) probiotic. The levels of TNF-α mRNA were not influenced by probiotic supplementation. Increases, although not significant, in the hematological and immunological parameters were also recorded. The data collected in the present study suggests that an LSB-supplemented diet acts on the gut immune system of sea bass by modulating the expression of the key inflammatory genes.

## 1. Introduction

Probiotics, according to the definition currently adopted by the Food and Agriculture Organization of the United Nations (FAO), are “live microorganisms which, when administered in adequate amounts, confer a health benefit on the host” [1,2,3,4]. Probiotics are opening a new era in health management strategies in aquaculture [4]. They are in fact well known as supplements in human diets [5,6], and they have recently gained attention in animal-rearing practices, because they are considered a functional food, even for aquatic species [4,7,8].

Dietary supplementation with probiotics offers many advantages in terms of both environmental sustainability and production. It allows the limitations and side effects of antibiotics or other drugs to be overcome, resulting in enhanced growth and disease prevention [2,4,9]. Probiotics affect certain elements of the non-specific immune system (such as monocytes, macrophages, neutrophils, natural killer cells, etc.). The use of probiotics could allow the use of antibiotics in aquaculture to be reduced, and an eco-friendly methodology to be generated to control pathogens, through the promotion of both the innate [10] and subsequentially adaptative immunity systems. Probiotic supplementation in aquatic species is known to enhance stress tolerance [11,12], disease prevention [11,13,14], and growth, as previously documented in rainbow trout [15], Nile tilapia [13], common carp [16], and channel catfish [17]. Numerous studies have been conducted involving the use of probiotics in aquaculture [7,15,18,19,20], and some of their mechanisms of action have been reported for the majority of probiotic strains. These mechanisms include effects on the feed conversion ratio [21,22,23,24], protection against pathogens through competitive exclusion of adhesion sites [25], the production of organic acids, hydrogen peroxide, antibiotics, bacteriocins, siderophores, lysozyme [3,7,12,16,23,26,27], and also the modulation of physiological and immunological responses [28,29,30].

However, in fact, it is still necessary to consider other variables, such as selective ingestion, manipulation in the intestinal tract, microbial interactions, and/or nutritional environment. The effect of probiotics can vary, according to several variables, including the form of supplementation (whole cells or cellular components, cell-free supernatants; encapsulated or freeze-dried probiotics), the level of dosage, and the duration of application, as well as the length of the delivery period [31].

The gut is the organ in which probiotics establish and perform their functions, including immunostimulatory activity [32,33]. Therefore, the interaction between probiotics, epithelial cells, and the gut immune system is of high interest to ensure fish health. Cytokines are one of the key regulators that modulate the immune response in fish [34] and it has also been shown that probiotics are actively involved in triggering strong responses by means of these chemical messengers. For example, it has been observed that fish fed with probiotics show an increased expression of immune-related genes, such as *il-1β*; *il-6*; *il-21*; *tnf-α*; and *tlr-1*, 3, and 4 [35]. The immunomodulatory properties of probiotics have only recently been explored in teleost [36,37,38] because the importance of the effects of probiotics on the immune response [39] and on the metabolism, which is reflected in the blood parameters, is becoming evident [40].

Numerous investigations have evaluated the sustainability and efficacy of feed supplementation, but the most well-documented probiotics are lactic acid bacteria (LAB) and *Bacillus* spp. [18,28]. However, yeast-based probiotics are also stimulating a great deal of interest [9,41,42] because of their ability to produce polyamines that participate in numerous biological processes [3,32,43]. Yeasts are active in different ways in the defense against pathogens: they adhere to and colonize the intestinal tract and produce active molecules, superoxide dismutase, and polyamine [3,44]; they act on epithelial cells through the stimulation of innate immunity receptors (Pattern Recognition Receptor, PRR), which are crucial for the key functions of the innate immune system [10]. PRRs recognize Pathogen-Associated Molecular Patterns induced by a cytokine release that is activated by leukocytes. IL-1β is produced in response to bacterial infections, while IL-6 is produced by macrophages and endothelial cells as a result of injury or trauma, and they represent both a pro-inflammatory mediator that stimulates B2 cells and a direct anti-inflammatory agent that triggers acute inflammation [45].

*Saccharomyces cerevisiae* var. *boulardii* (*S. boulardii*) has a prominent place among the most studied and characterized yeasts. From a taxonomical point of view, it is a yeast that belongs to the *S. cerevisiae* species, and it is used worldwide for the prevention and treatment of a variety of gut inflammatory diseases in both humans and animals [46,47,48,49,50,51,52], although very few data are available on its effects on aquatic organisms [53]. The molecular basis for the beneficial effects of *S. boulardii* was extensively studied both in vitro and in animal models, and it was found to have an impact on inflammatory cytokine production by intestinal epithelial cells [50,54].

Several studies have assessed the effects of probiotics on the immune status and gut microbiota of European sea bass, *Dicentrarchus labrax* (Linnaeus, 1758). The investigated aspects range from antioxidant and digestive enzyme activities, growth performances, gut histology and immunohistochemistry, and gut microbiota, to disease resistance and stress tolerance, the immunological and hematological response, body composition, malformations, and the survival rate [8,20,21,55,56]. However, there is a lack in the literature on the use of *S. boulardii* and its effects on the intestinal immune system of fish. In addition, the species/strain/stage specificity of probiotics is a topic of great concern, which needs to be addressed carefully and promptly. Indeed, the variability of literature data confirms the difficulty and complexity of elucidating the mechanisms of action as well as the benefits for the host, even in consideration of further differences that may be observed when similar probiotic species (i.e., members of the same genus) are used in the same fish species [57,58].

The aim of this study has been to investigate the effects of increasing the dietary supplementation of *Saccharomyces cerevisiae* var. *boulardii* yeast (LSB, LEVUCELL SB20^®^ TITAN, Lallemand Sas, Blagnac, France) on the welfare and health status of juvenile seabass reared in experimental aquaculture tanks. The intestinal cytokine expression, non-specific immune response parameters (lysozyme, spontaneous hemolytic activity SH50, hemagglutinating activity, respiratory burst), the antibacterial activities of the sera, skin, and intestinal mucus, and the peroxidase activity were all measured over a period of 90 days, to obtain insights into the potential beneficial effects of this yeast as a probiotic.

## 2. Materials and Methods

### 2.1. Ethics Statements

The experimental procedures were conducted in compliance with European Directive 86/609/EEC, which regulates the use of animals in research. The experiment was carried out from May/June to August 2013 at the Aquaculture Experimental Plant of the Institute for Marine Biological Resources and Biotechnology (IRBIM) (previously denominated IAMC, Institute for Coastal Marine Environment), CNR Messina (Italy). Fish were kept in indoor tanks equipped with a flow-through seawater system (with complete water renewal every hour). The photoperiod and thermoperiod were natural, and the photoperiod was approximately 14:L/10:D during the June and July months, while it was 13:L/11:D in August, and the water temperature ranged from 15.6 °C to 26.6 °C. The quality parameters of the water were maintained over a suitable range for European sea bass (36.0 ± 0.5 g L^−1^ salinity; 8 ± 1 mg L^−1^ oxygen) [59]. Euthanasia was conducted, previa sedation (0.25 mg/L), through the administration of a lethal dose (0.5 g/L) of Ethyl 3-aminobenzoate methanesulfonate (MS-222—Sigma Aldrich, St. Louis, MI, USA). After confirming death, that is, as the total loss of opercular movement, the fish were left in the euthanasia solution for at least 10 min.

### 2.2. Fish Rearing Conditions

European sea bass juveniles were obtained from a commercial farm in Sicily and they were acclimated to the new rearing conditions for three weeks before the onset of the trial. During this period, the fish were fed ad libitum on a control diet (Table 1).

The feeding experiment was conducted in twelve 1.4 m^3^ seawater fiberglass tanks with a continuous flow of seawater (24 changes per day, one per hour). Each tank was randomly stocked with 51 fish (average body weight of 174.6 ± 6.4 g; mean total length 24.5 ± 1.8), and each treatment was conducted in triplicate. Feed palatability was checked during the acclimation period, and the experiment was started when the fish began to consume the entire daily ration.

### 2.3. Diets and Feeding Regime

The *Saccharomyces cerevisiae* var. *boulardii* yeast (LSB, LEVUCELL SB20^®^ TITAN) was obtained from Lallemand Animal Nutrition (Blagnac, France). LEVUCELL SB contains a natural and selected live yeast, that is, the CNCM I-1079 strain (Pasteur Institute, Paris, France), which is metabolically active and in a micro-encapsulated form (Titan Technology).

Three isoproteic (50% crude protein) and isoenergetic diets (22.4 MJ kg^−1^) were obtained utilizing a basal diet that was formulated to meet the nutritional requirements of sea bass. The ingredients and composition of the basal diet are shown in Table 1.

After a preliminary test, performed on the same product, the diets were supplemented with LSB (2.0 × 10^10^ colony forming units (CFU)/g) at concentrations of 0 (LSB0, used as a control), 100 (LSB100), and 300 (LSB300) mg/kg of diet, respectively. After pelleting, one sample of each diet (500 g) was collected and analyzed. The yeast concentration corresponded to 5.4 × 10^5^ and 10.8 × 10^5^ CFU/g for the experimental groups fed with the LSB100 and LSB300 diets, respectively.

The experimental diets were distributed by hand, 6 days per week, twice a day (at 9:00 and 16:00) for 90 days (from June to August).

During the experimental period, the daily feeding rate ranged from 1.2 to 1.7% tank biomass per day^−1^, according to the water temperature in the tank. The water temperature ranged from 15.6 °C to 26.6 °C throughout the experimental period. The fish biomass present in each tank was determined each month by weighing fish in bulk after a 24 h period of starvation, in order to update their daily feeding rate.

### 2.4. Viability Test of the Probiotic in the Intestinal Tract

The fecal pellets of the reared sea bass were collected from each experimental tank during the weighing of the fish to verify the viability of the probiotic and its concentration in the intestinal tract. The fecal material was collected, using a sterile spatula, taking care to prevent contamination and to maintain aseptic conditions. Decimal (*w*/*v*) dilutions of the feces were prepared in 2% peptone water, with a pH of 6.5. Dilutions corresponding to final concentrations of 10^−8^, 10^−7^, and 10^−6^ were spread on the surface of Sabouraud Glucose Agar plates containing chloramphenicol (Fluka) and were incubated at 37 °C for 48 h. The colonies with typical morphological characteristics (white and smooth) were counted, and the counts were expressed as the number of Colony Forming Units (CFU) per gram of feces, considering the initial dilution of the intestinal contents.

### 2.5. Fish Sampling and Dissection

The fish sampling was conducted after 21, 45, and 90 days of dietary supplementation with the probiotic. Six fish from each feeding group (two fish per group) were randomly collected at each sampling time from their rearing tanks and were sacrificed with a lethal dose of anesthetic (see the Ethical Statement). The blood was immediately withdrawn from the caudal vein. Small aliquots of whole blood were collected in heparinized tubes for the respiratory burst activity measurements, while unheparinized blood samples were allowed to clot at 4 °C and centrifuged (1500 rpm for 10 min) for the other innate immune response parameters (lysozyme, hemolysins and hemagglutinins, peroxidase and bactericidal activities), and the obtained serum was stored at −20 °C for further analyses.

After blood sampling, the zootechnical parameters of each fish were measured, and this was followed by the collection of mucus and kidneys. A significant number of fish (*n* = 5) were used for the collection of skin and intestinal mucus. Skin mucus was removed by delicately scraping the skin with a sterile spatula, and it was stored in Eppendorf tubes at −20 °C for further lysozyme measurements and for the antibacterial activity assay against bacterial pathogens. Some reference strains—provided by the BIOMORF Dept., at The University of Messina (Prof. Laganà)—were used for this latter assay. *Escherichia coli* and *Pseudomonas aeruginosa* were chosen as Gram-negative bacterial representatives, while *Staphylococcus aureus* and *Staphylococcus epidermidis* were selected from the Gram-positive strains. Such pathogens were chosen as they are those used the most frequently in the antibacterial assays performed at the CNR laboratory, because of their clinical relevance; moreover, no fish pathogens (such as *Listonella anguillarum* or *Photobacterium damselae* subsp. *piscicida* or *Aeromonas hydrophila*) were tested in this assay, as previous trials performed on sea bass mucus had not given any positive results for these pathogens (unpublished data), and these species were therefore not included in our antibacterial assay. Finally, the whole intestinal tract of each fish was removed and separated into two parts on the basis of the visually different diameters: the anterior intestine (AI), that is, the tract following the pyloric ceca, and the posterior intestine (PI), immediately following the pre- and post-ileorectal valve, including the rectum. Samples were stored in RNA at 4 °C until RNA extraction. Intestine samples were stored in an appropriate fixative liquid for optical and electron microscopy analyses (see Section 2.7 and Section 2.8). Some whole intestines were used, together with skin and kidney samples, to determine the antibacterial properties of the intestinal mucus. 

For this analysis, intestine portions were dissected and incised with a sterile scalpel, and the mucus was gently scraped with a sterile spatula and stored in Eppendorf tubes at −20 °C until the assay was performed as described below.

The spleen and cephalic kidneys were collected and treated for extraction of the RNA and the lysozyme assay, respectively. The kidneys were weighed, and a 50 mg aliquot was homogenized in 100 µL of 0.1 M NaCl and the obtained homogenate was stored at −20 °C until analysis.

### 2.6. Zootechnical Parameters

Performance indexes were calculated to verify the effectiveness of the experimental diets on the productive performance of the fish, whereby the weights of the fish and their organs were measured before their treatment for the other analyses. The weight (BW, Body weight) and length (BL, Body Length) of the total fish and the weight of its intestine and visceral organs (stomach, liver, spleen, intestine, gallbladder) were measured to calculate the zootechnical parameters as follows.

iWeight gain (WG) (g): BW final (g) − BW initial (g)iiSpecific Growth Rate (SGR) (%): [(ln BW final (g) − ln BW initial (g))/feeding days] × 100iiiFeed Conversion Ratio (FCR): total supplied feed as DM (g)/WG (g), (where DM indicates Dry Matter)ivProtein Efficiency Ratio (PER): WG (g)/total protein fed (g)vFeeding rate (FR) (%): (ingested feed (g)/days of feeding) × 100

While the morphometric indices were calculated using the formulae:i.Viscerosomatic index (VSI): (viscera weight/BW) × 100iiHepatosomatic index (HSI): (liver weight/BW) × 100iiiCoefficient of fatness (CF): (perivisceral fat weight/BW) × 100ivCondition factor (K): (BW/BL^3^) × 100vRelative Intestinal Length (RIL): intestinal length/BW

### 2.7. Histological Analysis: Light Microscopy

Tissue samples were fixed in Bouin’s solution (Bio-Optica) (pH 1.75 at room temperature) for 48 h, processed for paraffin embedding, and cut into thin sections (3–5 μm) for microscopic observation. Sections were stained with hematoxylin-eosin (H&E) and examined using a light microscopy (Leica DMR) equipped with a camera (DFC295) and with image management software (Leica IM1000).

Histological analyses were carried out to evaluate the following intestinal morphological indexes: villi length (VL); villi thickness (VT); microvilli length (MT); number of goblet cells (GC); thickness of the lamina propria (LP), serous layer (SL), muscular layer (ML), and submucosa layer (SML).

The measurements and observations adopted in this study were based on a combination of criteria previously suggested by different authors [60,61], taking into account the possible effects of the utilization of a novel fish feed nutrient/additive on the gut and liver histology.

The criteria adopted in [62] were used for the liver morphology evaluation, as well as the quantification of the hepatocytes, to observe any possible alterations of the hepatocyte condition or hepatic cell morphology. The number of hepatocytes per area was quantified (125.000 μm^2^) [61].

### 2.8. Morphological Analysis: Scanning Electron Microscopy (SEM)

Intestine samples were fixed in a solution of 4.5% paraformaldehyde and 2.2% glutaraldehyde 0.1 M in cacodylate buffer with 5% sucrose (pH 7.5) and postfixed in 1% osmium tetroxide in 0.1 M cacodylate buffer with 5% sucrose for SEM analysis. The samples were then dehydrated in alcohol, critical point dried using liquid argon (Balzers CPD 030), coated with 3 nm gold-palladium in an SCD050 sputter-coated device (BAL-TEC), and examined under a Cambridge Stereoscan 240 SEM operating at 20 kV.

### 2.9. Innate Immunity Response Parameters

#### 2.9.1. Lysozyme Content in the Mucus, Plasma, and Kidney Samples

The lysozyme content was measured in the mucus, plasma, and kidney samples, by means of the diffusion method, on agarose plates containing a 0.05% suspension of *Micrococcus lisodeikticus* [63]. Aliquots (100 µL) of each sample were dispensed into wells produced in the thickness of the agarose; after incubation at 30 °C for 22 h, the diameter of the lysis halo obtained on the plates was measured and converted into Units/mL, through calibration with increasing concentrations of chicken egg white lysozyme (Sigma-Aldrich, St. Louis, MO, USA), which was used as the standard.

#### 2.9.2. Spontaneous Hemolytic Activity (SH50)

The method outlined in [64] was adopted to determine the Spontaneous Hemolytic activity (SH50) of the serum. Two-fold serial dilutions of serum were prepared in phosphate-buffered saline (PBS) and incubated at 37 °C for 1 h with a 2.5% sheep erythrocyte suspension in PBS. After centrifugation, the supernatant was collected, and its absorbance was measured at 540 nm using a Cary Varian UV/VIS spectrophotometer. The results were expressed as Spontaneous Hemolysis SH50 (SH50), in Units/mL, which was calculated from the reciprocal of the dilution at which 50% hemolysis was recorded, according to [65].

#### 2.9.3. Hemagglutininating Titer

The hemagglutinating activity of the serum was measured on mucus volumes (2 mL) from a pool of 3 individuals, which were centrifuged (2000 rpm) to remove scales or other skin debris that could have been collected during the sampling. The assay was performed on 96-well microtiter U plates (Nunc Inc., Roskilde, Denmark) through incubation of serial two-fold dilutions in PBS, of 20 µL of serum with an equal volume of a 2.5% sheep erythrocyte suspension, left at 35 °C for 1 h and at +4 °C overnight [66]. The obtained results were reported as the hemagglutinating titer, i.e., the reciprocal of the highest dilution that showed a visible agglutination of the red blood cells.

#### 2.9.4. Respiratory Burst Activity

The respiratory burst activity was assayed using a method based on the reduction of the substrate ferricytochrome C (Sigma-Aldrich) [67]. The reduced cytochrome C was determined spectrophotometrically on the basis of the difference in the absorbance readings between 550 and 468 nm, which is proportional to the amount of reduced cytochrome C. Controls were also included in the reaction, using the Superoxide Dismutase enzyme (SOD, 300 IU ml^−1^ Sigma-Aldrich), which prevents the reduction of cytochrome C and of the Zymosan A (Sigma-Aldrich) and 4-a Phorbol 12-myristate 13-acetate (PMA, 1 µg ml^−1^ Sigma-Aldrich) substrates, which stimulate respiratory burst activity; incubation was performed at 22 °C for 30 min. Triplicate measurements were performed in the assay.

#### 2.9.5. Antibacterial Activity of the Serum, Kidneys, and Skin and Intestinal Mucus

Serum, intestinal, skin mucus, and kidney samples were assayed for their possible antibacterial activity against certain reference strains of bacterial pathogens. An analytical procedure that was reported in a previous study [68] was followed. An axenic culture with a final concentration of 10^8^ cells/mL in Brain Heart Infusion broth—as assessed through a comparison with a 0.5 McFarland standard of turbidity—was prepared for each bacterial pathogen. A 4 mm thick aliquot of 1% NaC1 was added to the surface of Tryptic Soy Agar (Oxoid) plates, according to the standard Kirby–Bauer procedure, and was inoculated with 0.1 mL of each bacterial culture, using a sterile cotton swab. Disks of 6 mm diameter sterile pads were placed on the plates and each one was soaked with 20 μL of sample (serum, skin or intestinal mucus, or kidney homogenate). After incubation of the plates at 35 °C for 24 h, the diameters of the inhibition zones of bacterial growth were measured using a precision caliper (Mitutoyo, Andover, UK).

#### 2.9.6. Peroxidase Activity of the Serum

The peroxidase activity of the serum was measured according to the method outlined in [69]. Briefly, 30 μL of serum was diluted with 120 μL of HBSS buffer without Ca^2+^ or Mg^2+^ on flat-bottom multi-well (96 wells) plates. Thereafter, 50 μL of a 10 mM solution of tetramethylbenzidine chloride (TMB, Sigma-Aldrich) and 5 mM hydrogen peroxide were added as substrates. The change in color of the reaction mixture was stopped after 2 min of incubation through the addition of 50 µL of 2 M sulfuric acid. The absorbance of the reaction product was measured at 450 nm using a spectrophotometer equipped with a 96-well plate reader (Microtiter reader ELx-808, Bio Whittaker Europe, Walkersville, MD, USA). Blanks were prepared with the reaction mixture but without the sample, as controls.

### 2.10. Analysis of Cytokine Gene Expression

#### 2.10.1. RNA Extraction and cDNA Synthesis

The total RNA was extracted from the proximal and distal tracts of the gut (n = 6 for each experimental group; approximately 90 mg per sample) using TRIzol reagents (Invitrogen, Waltham, MA, USA), according to the manufacturer’s instructions. NanoPhotometer^®^ P-Class (Implen, Munich, Germany) and Gel Red™ (Sigma Aldrich, Milan, Italy) staining of the 28S and 18S ribosomal RNA bands on 1% agarose gel was used to assess the RNA concentration and integrity.

Complementary DNA (cDNA) was synthesized from 2 µg of total RNA using a High-Capacity cDNA Reverse Transcription Kit (Applied Biosystem, Waltham, MA, USA) in a 20 μL reaction containing 2X Reverse Transcription Master Mix and MultiScribe Reverse Transcriptase; cDNA was then diluted 1:10 in RNase-DNase-free water. An aliquot of cDNA was used to check the specificity of the primer pair.

#### 2.10.2. Quantitative Real-Time PCR

The gene expression of interleukin-1 (*il-1β*), interleukin-6 (*il-6*), and tumor necrosis factor alpha (tnf-α) were assessed using real-time PCR. ß-actin was used as a housekeeping gene.

The primer sequences, melting temperature, and Gene Bank Accession number are shown in Table 2. Primers were designed on the basis of the genomic sequence deposited in Gene Bank. The qRT-PCR reactions were performed by means of a TaqMan Universal PCR Master Mix (Applied Biosystems, Foster City, CA, USA), using 5 ρmol of each primer and 9 ρmol of both the target and beta-actin probes.

Negative controls revealed no amplification of the product, and no primer-dimer formation was found in the control templates. PCRs were performed in triplicate using a 7300 PCR real-time System (Applied Biosystems), setting the reaction as follows: 3 min at 95 °C, 45 cycles of 20 s at 95 °C, 20 s at the annealing temperature (see Table 2), and 20 s at 72 °C. A fluorescent signal was detected at the end of each cycle, and melting curve analysis was performed to confirm that only one PCR product was present in these reactions. The expression level of the analyzed genes was quantified using the comparative Ct method (2^−ΔΔCt^) and expressed as the ± n-fold difference versus the standard.

### 2.11. Statistical Analysis

Measurements were performed in triplicate for each parameter, and the results were reported as the mean value ± standard deviation (SD). The occurrence of significant differences among the experimental groups was evaluated using one-way analysis of variance (ANOVA), which was performed on the obtained dataset. Before ANOVA, the assumption of normality of data and homogeneity of variance was tested using one-way Kolmogorov–Smirnov (K–S) and Levene’s tests, respectively.

As a consequence of the non-normality of the dataset, the values of all the analyses were log-transformed [ln(x + 1)]. The Kruskal–Wallis test was used for heterogeneous variances. PAST^®^ software (version 4.0) was used for the statistical analysis in order to test differences among the diets for each parameter at a single timepoint; *p* values < 0.05 were considered statistically significant.

## 3. Results

### 3.1. Viability of the Probiotic

The concentrations of yeast present in the feces and still viable on the Sabouraud Glucose Agar selective medium (Appendix A) showed values that on average were of the order of 10^4^ Colony Forming Units (CFU)/g, ranging from 7.80 × 10^3^ to 4.30 × 10^4^ CFU/g in the sea bass fed with LSB 100, and from 1.73 × 10^4^ to 1.03 × 10^5^ CFU/g in the fish fed LSB 300 (Appendix A). As expected, the abundance of yeast in the intestine increased over time, peaking at T90, but no significant differences were found between the groups of sea bass fed with a probiotic diet of 100 and 300 mg/kg feed (F = 4.01), or for the different times (F = 4.23).

### 3.2. Zootechnical Parameters and Morphometric Indices

The results of the zootechnical parameters and morphometric indices are shown in Table 3. Differences among the experimental groups were statistically analyzed for each sampling using the one-way ANOVA test, considering the diet as a discriminating factor. The results showed that there were no significant differences for any of the zootechnical or morphometric indices.

### 3.3. Histological Analysis: Microscopy

The analysis of the histological parameters of the proximal gut (Table 4) revealed statistically significant differences in villi length (VL) between the three treatments. A significant (*p* < 0.05) elongation of the length of the villi was observed in the LSB300 group, compared with the LSB100 group, after feeding for 21 days. Statistical differences were also observed in the SML thickness (SML). The SML thickness of the LSB100 group was significantly higher than the control after 45 days of feeding. The length of the microvilli showed significant differences between the probiotic-supplemented diets (LSB100 and LSB300) and the control after feeding for 90 days.

The analyses on the distal portion of the intestine did not show any significant differences, except the presence of yeast colonizing the mucosa detected with SEM.

The histological study of the liver, which considered the number of hepatocytes per area, was performed as described in Section 2.7. Our results showed the occurrence of significant differences between the control group and both the LSB100 and LSB300 treatments. Significant differences were in particular found between the control and the LSB100 group after 21 and 90 days of feeding, while the differences between the control and the LSB 300 group were significant for all the sampling times (after 21, 45, and 90 days of feeding). Overall, the greatest difference in the hepatocyte numbers, in relation to the supplementation with the probiotic, was found in the fish fed the diet with the highest concentration of *S. cerevisiae* var. *boulardii*.

### 3.4. Scanning Electron Microscopy (SEM)

The colonization of yeast in the intestinal mucosa was evident from the SEM results. As can be seen from the comparison of the two intestinal tracts photographed with the SEM, *S. cerevisiae* was absent in the anterior intestine but it was instead concentrated in the posterior tract (Appendix A).

### 3.5. Immune Innate Response

The results regarding all the immune innate response parameters are reported in Table 5.

#### 3.5.1. Lysozyme

The lysozyme content reached higher values in the skin mucus, plasma, and kidneys of the sea bass fed diets containing an LSB300 dietary supplementation, as well as in those fed an LSB100 supplementation, albeit only for the skin mucus and plasma. However, the quantitative differences recorded among the experimental groups were not statistically significant. Similar patterns were observed during the experiment; higher lysozyme values were measured 45 days after feeding with the probiotic supplementation. Prolonging feeding to 90 days resulted in slightly lower or constant values in the skin mucus or plasma, or in the kidneys, respectively. The LSB300 dietary supplementation induced an increased immune response in this organ that was on average 1.12 times higher than the control values.

#### 3.5.2. Hemolytic Activity (SH50)

The hemolytic activity values increased in the fish fed with probiotic-supplemented diets, with a greater response in the group of sea bass fed with the highest probiotic dose (LSB300), where the SH50 values were 1.72 and 2.27 times higher than the control values after 21 and 45 days of dietary supplementation with LSB, respectively. However, the differences between the three experimental groups were not significant. After 45 days of feeding, a slight decrease was recorded in the hemolytic activity values, which then increased after 90 days.

#### 3.5.3. Hemagglutinating Titer

The effect of LSB dietary supplementation on the hemagglutinating titer values was already evident after 21 days of feeding for the highest dose of *S. cerevisiae* var. *boulardii*, although the increase was not significant. On the other hand, contrasting results, characterized by the absence of variations (at T45) or by opposite trends to the previous ones, i.e., with a depressive effect on the hemagglutinating activity for an increase in the probiotic concentration, and a peak in the control group were observed for the successive samplings.

#### 3.5.4. Respiratory Burst Activity

Respiratory burst activity, which was measured as the production of superoxide anion, highlighted that LSB induced a stimulating effect on the sea bass fed the LSB300-supplemented diet, with values that increased 4 and 11 times compared to the control values at 45 and 90 days after feeding with LSB. This effect, although not significant, appeared to be delayed, compared to the other parameters of the immune response.

#### 3.5.5. Peroxidase Activity

The peroxidase activity values measured in the serum depicted a variable pattern, which initially showed an insignificant increase in the sea bass fed with the diet containing LSB 100 and reached almost triplicate values (>0.24 U/mL), after feeding for 90 days, in those fed with LSB 300.

#### 3.5.6. Antibacterial Activity of the Serum, Kidneys, Skin, and Intestinal Mucus

The results of the assay on the antibacterial activity of the serum, kidneys, intestinal, and skin mucus against the target pathogenic bacteria were expressed in terms of the percentage of the total number of examined samples that showed a positive reaction (Table 6).

The serum samples showed antibacterial activity almost exclusively against *S. epidermidis*; only after feeding for 90 days was a positive reaction against *P. aeruginosa* detected in the LSB300 group. No activity was observed against *E. coli* or *S. aureus*.

High percentages of the kidney samples showed antibacterial activity toward P. aeruginosa. A low percentage of samples also exhibited activity against *E. coli*, while no activity against *S. epidermidis* was observed. A high percentage of samples was positive after feeding for 21 days, even in the control group; no clear correlation was observed between the number of positive reactions and the dose of the probiotic contained in the diet at T45 or T90.

A high antibacterial activity against all the tested bacteria was observed for all the skin mucus samples. Greater antibacterial activity was recorded for T21, T45, and T90, probably in relation to the higher temperatures of the summer months than those of the spring season (T0 sampling), while no differences were found among the various experimental groups.

The intestinal mucus samples showed a high bactericidal activity against all the tested pathogens, and in particular against *P. aeruginosa* and *S. epidermidis*, while the activity against *E. coli* was lower. A positive response was not found for *S. aureus*, except for the control group after feeding for 45 days. In general, there was more activity at 90 days; no significant relationship was highlighted, in relation to the diet, between the expression of the antibacterial activity and the levels of probiotic supplementation.

### 3.6. Cytokine Gene Expression

The *il-1β*, *il-6*, and *tnf-α* genes were examined to evaluate whether the probiotic treatment had an effect, at the gut mucosal level, on cytokine mRNA expression, using a real-time PCR method. The expression of the cytokine genes is reported in Table 7 (proximal gut) and Table 8 (distal gut). There were no significant differences in *il-1β* expression in the proximal gut between the probiotic-supplemented groups and the control, at any of the sampling times, whereas the level of *il-1β* mRNA was significantly higher in the LSB100 group after 45 days than in the LSB300 group. Instead, the level of *il-1β* mRNA was significantly lower in the distal portion of the gut in the LSB100 group after 21 days of feeding.

The gene expression level in the proximal gut of *il-6* was down-regulated in both probiotic groups. After 21 days of feeding, the LSB100 and LSB300 groups both showed a significant difference from the control and between each other. At this time of sampling, the expression of the *il-6* gene was significantly lower in the fish fed the LSB300 probiotic. After 90 days, the expression of the *il-6* gene in LSB300 was significantly higher than in the control and LSB100 groups. No significant differences in expression were observed for the distal gut, except after 45 days, when the LSB100 *il-6* gene expression was significantly higher than the LSB300 group. The levels of TNF-α mRNA do not seem to have been influenced by probiotic supplementation. No significant differences between groups were observed, either over time or in the different portions of the gut.

## 4. Discussion

The comprehensive dataset collected in our study can be considered a first contribution to knowledge on the effects of dietary supplementation with *S. cerevisiae* var. *boulardii* in sea bass. Although the beneficial effects of diets supplemented with this yeast species have been documented for mammals [50,70], and have recently been reviewed in several animal models [71], the potential effects on immune system parameters are yet unknown for fish, and only a few reports have documented the effects on the growth parameters of some freshwater fish species [72].

Although several studies have reported on the use of probiotics in aquaculture nutrition, caution should be taken in the interpretation of the biological data, because both zootechnical parameters and morphometric indices can vary to a great extent in relation to different variables, such as the stocking density [73], and no significant differences might be observed in the case of enriched diets [74].

As far as the biometric measurements are concerned, none of the zootechnical parameters or morphometric indices examined in our study showed significant differences related to the administered experimental diets, as expected, considering the results published in the literature. The supplementation with *S. cerevisiae* var. *boulardii* had positive effects on the growth performance, health, and survival rate of the reared fish, especially when the probiotic was added at a concentration of 2.5% of body weight [72]. An enhancement of growth performance and zootechnical parameters has been reported for fish fed with *S. cerevisiae* enriched diets [11,13,75]. Finally, positive effects on growth performance were registered for sea bass fed with *S. cerevisiae* administered at a dose of 15 × 10^9^ CFU/g, thus suggesting that the response to administration could change depending on the yeast concentration [40]. A significant difference has also been reported for this species when testing different strains of yeast [8].

The dietary inclusion of *S. cerevisiae* has been shown to accelerate digestive system maturation and increase nutrient digestibility as a result of improvements in the integrity of the intestinal mucosa and the density of the intestine villi [11].

Nile tilapia fed with probiotics, in particular *S. cerevisiae*, have shown a significant increase in villi length [13,43,75]. This elongation of the villi was related to the absorption surface and it occurred above all in the anterior part of the intestine [75]. Such findings are consistent with our results, where there was a significant difference in villi length, especially in the proximal (or anterior) part of the intestine. The elongation of the villi was proportional to the probiotic concentration after feeding for 21 days, while, after 90 days, the longest length was recorded in the diet with a lower concentration of the LSB100 probiotic. The explained changes in absorption surface are usually matched with a consistent increase in the thickness of the epithelium layers [43], thus supporting our finding regarding the submucosal layer (SML) thickness.

In addition, in our study, the microvilli thickness was found to increase, which resulted in an improved absorption, without any events of necrosis or cell disruption, thus confirming the results obtained for the same species by Rawling et al. [8].

The histological examination results are in agreement with previous results on the same species [53], where the liver analysis showed the greatest difference between the control and the LSB300 group for all the sampling times. A significant increase in the number of hepatocytes-area was detected, thereby showing a positive effect of the probiotic on the fish, especially for the highest concentration [76]. Conversely, none of the negative signals related to the supplementation of alternative diets reported in the literature were observed [77]; in fact, none of the experimental groups showed high vacuolization scores, thus indicating there were no signs of steatosis in the trial.

Nutritional status is recognized as an important indicator of the resistance of fish to diseases [78]; the relationships that link nutrition to the immune response in fish have been widely investigated [4,15,16,79,80], and fish diseases have been found to occur more frequently in fish stressed by different factors, e.g., a sub-optimal temperature [39].

Among the beneficial effects of the use of probiotics in fish, those related to the modulation of the immune system are well known. Probiotics are effective in modulating the inflammatory response and in increasing disease resistance to bacterial infections, as demonstrated in *Sander lucioperca* protected *by Lactobacillus brevis* against *Aeromonas hydrophila* [81]. Moreover, Maricchiolo et al. [53] demonstrated that *S. cerevisiae* var. *boulardii*, administered at a dosage of 300 mg/kg (after 21 days of feeding), is able to protect the intestinal mucosa of gilthead seabream against *V. anguillarum* infections.

Enhancement of the immune response following dietary supplementation with probiotics has been widely documented [42,82]. Probiotics are known to determine an increase in the activity of the complement system of fish [83,84], and yeasts have been reported to be capable of stimulating the immune response of these aquatic organisms. Among the various yeasts, *S. cerevisiae*, administrated as whole cell or cell walls, may provide very important non-nutritive compounds, including various immunostimulating compounds such as glucose polymers (β glucans), mannan oligosaccharides [85], nucleic acids and chitin, which may be beneficial to fish health [9,32].

*S. cerevisiae* var. *boulardii* has been reported to act as a pathogen antagonist and immunomodulator in mammals [50,70]; an increase in the production of secretory IgA after feeding with *S. cerevisiae* var. *boulardii* has also been reported in rats and mice [86,87]. *S. cerevisiae* is known to increase the growth performance, immune responses, and disease resistance of several species of fish, such as gilthead seabream [32,88,89], Nile tilapia [11,90], Channel catfish [17], and rainbow trout [91,92]. The beneficial effects of this yeast on growth, physiological responses, and gut microbiota have also been reported in beluga juveniles by Hoseinifar et al. [93] and in *D. labrax* [8].

The effects on the non-specific immune parameters of dietary supplementation with probiotics can in general be affected to a great extent by the fish species, the stage of development, the strain of the probiotic, dosage, feeding duration, feed composition, mode of supplementation and/or certain environmental-rearing conditions, such as water temperature fluctuations [94]. With respect to this latter variable, in our feeding trial, the temperature ranged over about 11 °C and such a thermic variability could have affected the response of non-specific immune parameters to the yeast administration, as already reported for other fish species [95]. The probiotic efficiency of *S. cerevisiae* seems to depend on the used strain [58], and a difference between single-strain and multi-strain probiotics has also been detected [8,19]. In addition, the viability of a probiotic is also important, as shown for rainbow trout, where the immune response of fish may vary according to the probiotic status, that is, active or inactivated [96,97].

In the present study on sea bass, the viability of the probiotic administered with the diet was assessed by evaluating the abundance of yeast in the feces; the amount of viable yeast was gradually increased during the experiment. However, only a study through electron microscopy can provide information on the type of yeast present in the gut, i.e., can reliably indicate whether the flora has colonized the intestinal mucosa transiently or actively. The presence of *S. cerevisiae* var. *boulardii* CNCM I-1079 was not detected in the gut of the juvenile sea bass, and a low level of colonization was found, even after 9 months; the same strain did not seem to affect the intestinal bacteria of rainbow trout.

Regarding the effects of *S. cerevisiae* var. *boulardii* on the lysozyme content, slight and insignificant increases in the values of this parameter were observed during the present study. Rodriguez et al. [98] reported significant increases of lysozyme in *Sparus aurata*, 2–4 weeks after dietary supplementation with a mutant strain (fks-1) of *S. cerevisiae* as a probiotic. Song et al. [99] recorded an increase in the activity of lysozyme and bactericidal activity in *Miichtys miiuy* fed a mixture of microorganisms, that is, both bacteria and yeasts (*Clostridium butyricum*, *Bacillus subtilis*, *Lactobacillus acidophilus*, *S. cerevisiae*), which would seem to suggest that stimulation of the immune system had occurred. Supplementation with *S. cerevisiae* var. *boulardii* CNCM I-3799 enhanced the lysozyme, the complement activities, and the Ig level of rainbow trout [58]. The enhancement of lysozyme activity due to the use of *S. cerevisiae* has also been reported in other species, such as *Pangasianodon hypophthalmus* [100]. The lysozyme activity was found to show a dose-dependent response, which was affected by *S. cerevisiae* supplementation of the diet [40].

An improved immunological response (serum lysozyme activity and total immunoglobulins) has recently been recorded for *Pangasianodon hypophthalmus* fingerlings fed for 8 weeks on diets supplemented with a mixture of yeast (*S. cerevisiae*) and bacteria (*Bacillus subtilis*, *Lactobacillus plantarum*, and *Enterococcus faecium)* [101]. Similar increases in the serum lysozyme content have also been found in largemouth bass (*Micropterus salmoides*) fed for 8 weeks with *Bacillus subtilis*, *Lactobacillus plantarum*, and *S. cerevisiae*) [102].

The increases in the hemolytic activity recorded in the examined sea bass, although not significant, were related to the beneficial effects produced by probiotics on the hematology parameters, which are commonly referred to as indicators of the physiological state of fish, in agreement with what Mohapatra et al. reported [103]. Chiu et al. [104] reported a significant increase in the activity of the complement (ACH50) and lysozyme content in grouper (*Epinephelus coioides*) fed a diet supplemented with *S. cerevisiae* at concentrations of 10^5^ and 10^7^ CFU/kg of body biomass, compared to the control diet (without any probiotic). A significant increase in ACH 50 activity (and total Ig and lysozyme activity) has been observed in striped catfish (*Pangasianodon hypophthalmus)* in the presence of *S. cerevisiae* enriched diets (10^6^ and 10^8^ CFU *S. cerevisiae* g^−1^ diets) after an experimental period of 120 days [100].

Chiu et al. [104] found that the alternative complement activity (ACH50), together with the phagocytic activity, respiratory burst, and serum lysozyme activity of *Epinephalus corioides* fed diets containing *S. cerevisiae* at 10^5^ and 10^7^ CFU/kg were significantly higher than those of fish fed a lower probiotic concentration (10^3^ CFU/kg).

Abu-Elala et al. [90] found a significant increase in the growth performance and stimulation of non-specific cellular/humoral immunological parameters, including lysozyme, phagocytosis, and cytokine production, an alternative complement pathway in cultured *Oreochromis niloticus* fed with different forms of *S. cerevisiae* (namely as a whole yeast cell, its extract (mannan-oligosaccharide) and as a mixture of the whole cell and its extract).

The patterns described for the hemagglutination titer in sea bass fed with *S. cerevisiae* var. *boulardii* were quite variable, and they therefore did not allow us to correlate any effect with the administration of the probiotic. Furthermore, the reduction in the hemagglutinating activity measured during the last sampling could also be associated with a depression in the activity of the complement, as previously observed by Rodriguez et al. [98] in *Sparus aurata* after 6 weeks of feeding with the *S. cerevisiae* mutant strain fks-1.

Sakai [105] reported a progressive and dose-dependent increase in the respiratory burst activity of tilapia, which has also been observed for the respiratory burst values observed in the sea bass during the present study, albeit not at levels of statistical significance. Increases in the respiratory burst and the intestinal lysozyme contents were also measured by Kim and Austin [106] in rainbow trout after administration of probiotics. Ortuño et al. [89] reported that the administration of *S. cerevisiae* had a positive effect on both the respiratory burst and the complement activities.

*S. cerevisiae* has been shown to have a marked effect on the respiratory burst activity of Nile tilapia, where the supplementation of 2, 4, 6, and 10 g of *S. cerevisiae*/kg diet was found to determine the linear and quadratic enhancement of respiratory burst and lysozyme as well as superoxide dismutase, catalase, and glutathione peroxidase [13]. The findings pertaining to rainbow trout reveal that the use of *S. cerevisiae* as a probiotic (10^7^ CFU/g) significantly affects the respiratory burst activity, especially after 130 days of treatment [107]. Harikrishnan et al. [108] found an immunostimulant and protective effect against infection with *Uronema marinum* in olive flounder (*Paralichthys olivaceus)* supplemented with *S. cerevisiae*, with a significant increase in lysozyme and superoxide anion production after 4–8 weeks of feeding. Significant improvements in respiratory burst activity have also been reported in tilapias fed diets supplemented with *Bacillus subtilis*, *S. cerevisiae* and *Aspergillus oryzae* as probiotics (*p* < 0.05), compared to control fish without any probiotics [94].

A single inclusion of *S. cerevisiae* in the diet of rainbow trout over 130 days led to increased respiratory burst activity and an increased number of blood cells (neutrophils, lymphocytes, and monocytes) [107].

Regarding the antibacterial activity of the serum, kidneys, and skin and intestine mucus, a discrete activity was found in sea bass fed with *S. cerevisiae*, particularly at the skin mucus level. However, the results of the antibacterial activity assay did not allow a clear stimulating action on the bactericidal activity following the probiotic supplementation to be observed. Conversely to our findings, Taoka et al. [109] reported increases in the antibacterial activity of plasma in *Oreochromis niloticus* after 15 and 30 days of feeding with such probiotics as *Bacillus subtilis*, *Lactobacillus acidophilus*, *Clostridium butyricum*, and *S. cerevisiae*. Moreover, significant antimicrobial activity against several pathogens, such as *E. coli*, *Shigella*, *Salmonella*, *V. cholerae*, *C. difficile*, and *Candida albicans*, was recorded following a dietary supplementation with *S. cerevisiae boulardii* CNCMI-745 in several animal models [54].

The patterns of peroxidase activity obtained during the present experiment suggest that *S. cerevisiae* had a stimulating, although somewhat delayed, effect on this parameter, as observed at the last sampling. This result was in contrast with the depression of peroxidase activity reported in the serum of *S. aurata* by Rodriguez et al. [98], 6 weeks after the administration of *S. cerevisiae* (mutant strain fks-1).

It has also been shown that peroxidase activity increases significantly in red sea bream (*Pagrus major*) fed with *S. cerevisiae* or *B. subtilis*, while, surprisingly, the catalase activity decreases [110]; the concentration of the probiotic can also affect the peroxidase activity in red seabream [18]. Moreover, the presence of *S. cerevisiae* also seemed to enhance the peroxidase activity in Atlantic salmon receiving dietary supplementation with *Kluyveromyces marxianus*. The same results were obtained in Nile tilapia (*Oreochromis niloticus*) fed diets supplemented with *S. cerevisiae* [13]. It has recently been demonstrated that *S. cerevisiae* enhances the peroxidase activity in European sea bass, with a simultaneous increase in the lysozyme and other antioxidant enzyme contents [40].

More in general, Tukmechi et al. [111] found that the dietary administration of *S. cerevisiae* stimulated the immune function and enhanced resistance to *Yersinia ruckeri* in *Oncorhynchus mykiss*. A similar result was obtained by Sheikhzadeh et al. [58]. Instead, Tewary and Patra [112] showed that the yeast cell wall enhanced the innate immunity and growth parameters of *Labeo rohita*, and also stimulated fish disease resistance; enhanced non-specific immune responses were also recorded by Bandyopadhyay et al. [113] in juveniles of this species.

Abu-Elala et al. [90] reported a significant improvement in the growth, performance, and non-specific cellular-humoral immunological parameters of another species, *Oreochromis niloticus*, fed a fully fermented diet as well as an enhancement of resistance against some fish pathogens.

In short, the differences in our experimental trial in the hematological and immune response parameters induced by the LSB administration were not significant. This result is consistent with the absence of the differences reported by Munir et al. [114] in snakehead (*Channa striata*) fingerlings after feeding with prebiotics and probiotics (*S. cerevisiae*, beta-glucan, galactooligosaccharides, mannan-oligosacchafrides, *Lactobacillus acidophilus*) for 16 weeks, followed by 8 weeks of feeding with a control diet without any pre- or probiotics supplementation.

In the present study, *S. boulardii* dietary supplementation significantly modulated the expression of inflammatory interleukin genes in the gut of sea bass. The considered dosages both had a positive effect on the expression of the interleukins in the gut, but the most evident benefit was observed after 21 days of feeding, especially in the proximal gut (see our histological results).

IL-1β is one of the first pro-inflammatory cytokines to have been expressed [115]. In our study, real-time PCR detected a lower *il-1β* transcript in the probiotic groups than in the controls after 21 days of feeding. This result is in further agreement with the literature that has reported the downregulation of IL-1*β* in *Micropterus salmoides* [102], *Epinephelus coioides* [116], and *Oncorhynchus mykiss* [117] when a diet is supplemented with yeasts.

Moreover, again regarding IL-6, our studies noted a downregulation of transcription at day 21 in the groups fed on a supplemented diet, in a dose-dependent manner, compared to the control group.

IL-6 is a potent, pleiotropic cytokine produced by various cells to regulate hematopoiesis, inflammation, immune responses, and bone homeostasis [45]. It plays a major role in the humoral immunity of fish and, under normal conditions, its level is low, while, in the course of pathogen infection, its transcription is upregulated in the spleen and kidney head [118,119]. Unfortunately, to the best of our knowledge, the expression of the *il-6* gene in the gut has never been explored in fish, especially under normal conditions in the absence of damage of any kind.

In our study, the down-regulation of both interleukins in response to the administration of *S. boulardii* could be interpreted as a positive signal of “gut health” in *D. labrax*.

TNF-α is an important component during the onset of early inflammatory events. TNF-α is synthesized by various cell types upon stimulation with endotoxin, inflammatory mediators, or cytokine, such as IL-1, and, in an autocrine manner, upon stimulation by TNF itself. In our study, the expression of *tnf-α* was not influenced by the administration of the probiotic. The behavior observed in TNF-α may be the result of a combination of factors, such as the organ in which the expression was analyzed, the fish species, and the administered probiotic.

*S. cerevisiae* usually determines the up-regulation of TNF- α, which is coupled with a histological variation, such as an increase in goblet cells in different species [42,90]. Our results are in agreement with this scenario, as none of the aforementioned signals appeared, thus suggesting that the doses used for probiotic addition do not determine any stimulation of this pro-inflammatory cytokine. Consequently, it is possible to suggest that the response of those genes could appear after a longer period than 90 days, as pointed out by Vazirzadeh et al. [107], or for higher concentrations of yeast (10^6^) [3,42].

In conclusion, as also documented for other animals, yeasts, and *S. boulardii* in particular, modulate the immunological function by decreasing the levels of kinases and pro-inflammatory molecules [120].

The present results may indicate that a similar immunomodulatory action takes place in the gut of *D. labrax*, thus suggesting a possible anti-inflammatory effect of this probiotic strain.

## 5. Conclusions

In conclusion, LSB was found to have a positive effect on *D. labrax* at the tested concentrations because it did not impair either the growth parameters and morphometric indexes, or the intestinal mucosa structure, and it positively modulated the expression of the key inflammatory genes of the sea bass gut immune system. LSB, at a dosage of 300 mg/kg, lowered the expression of the interleukin-6 gene, when it was administrated over short time periods. Indeed, Maricchiolo et al. [53] recommended a diet supplementation (with this dosage) for 21 days because of its marked protective effects on intestinal mucosa. On the contrary, we observed that IL-6 was upregulated after a prolongated administration. This behavior was confirmed by an increasing trend of the respiratory burst and hemolytic activity parameters, where a stimulating effect was observed after 90 days. Consequently, further studies in which the probiotic administration is extended to longer periods (>90 days) are recommended to draw robust conclusions on the administration of *S. cerevisiae* var*. boulardii* as a probiotic for sea bass.

## Figures and Tables

**Table 1 animals-13-03383-t001:** Ingredients and proximate composition of the basal diet expressed as a percentage of the total diet.

Ingredients (g/kg)	LSB0	LSB100	LSB300
Herring fish meal (CP ^1^, 70%)	480	480	480
Corn gluten	140	140	140
Gelatinized starch	135	135	135
Soybean meal (CP ^1^, 44%)	115	115	115
Cod liver oil	110	110	110
Vitamin mixture ^2^	10	10	10
Mineral mixture ^3^	10	10	10
LEVUCELL SB20^®^ TITAN	-	0.1	0.3
**Proximate composition, % DM ^4^**			
Dry Matter	94.5		
Crude protein	50.0		
Ether extract	15.3		
Ash	8.2		
Nitrogen free extracts ^5^	26.5		
Gross energy ^6^, MJ/kg DM	22.4		

^1^ CP = crude protein; ^2^ per kg of diet: Vit. A 200 IU; α-tocopheril acetate 16 mg; Niacin 72 mg; Vit. B6 16 mg; Choline 0.48 mg; ^3^ per kg of diet: Ca 22.60%; P 9.90%; Mg 7.20%; Na 1.70%; ^4^ DM = dry matter; ^5^ Calculated as: 100 − (% Crude protein +% Ether extract +% Ash); ^6^ Calculated as: [Crude protein (g) × 23.6 + Ether extract (g) × 39.5 + Nitrogen 3]/100.

**Table 2 animals-13-03383-t002:** Primers sequences used in real-time PCR. AT: annealing temperature.

Cytokine Gene	Primer Sequence	AT°	Accession No.
*il-1ß*	FW: GAGACACTGATGAGCACTGAGT RV: CTGATGTTCAAACCGGAGTC	61.2 62.2	AJ269472.1
*il-6*	FW: AAACATGCCCTGAGAAGTCC RV: TTGACGTGTTCTCTGTGCCT	63.0 63.6	AM490062.1
*tnf-α*	FW: CTCAACACAGCGGATATGGA RV: CCTTCTAAATGGATGGCTGC	63.7 63.4	DQ070246.1
ß-Actin	FW: GGTACCCATCTCCTGCTCCAA RV: GAGCGTGGCTACTCCTTCACC	69.0 61.9	AJ537421.1

**Table 3 animals-13-03383-t003:** Zootechnical parameters (mean ± standard deviation, *n* = 3) and morphometric indices (mean ± standard deviation, n = 6).

	Days of Feeding	LSB0	LSB100	LSB300
Weight Gain (g)	21	21.13 ± 3.77	22.77 ± 3.59	22.06 ± 5.94
	45	54.53 ± 2.45	56.00 ± 7.74	56.66 ± 8.71
	90	105.22 ± 8.56	105.24 ± 8.90	97.35 ± 13.36
Specific Growth Rate (%)	21	0.55 ± 0.11	0.61 ± 0.09	0.56 ± 0.13
	45	0.55 ± 0.02	0.57 ± 0.07	0.55 ± 0.05
	90	0.61 ± 0.05	0.62 ± 0.03	0.56 ± 0.04
Feed Conversion Rate	21	1.93 ± 0.24	1.86 ± 0.31	2.10 ± 0.59
	45	2.30 ± 0.11	2.24 ± 0.27	2.31 ± 0.21
	90	2.28 ± 0.19	2.26 ± 0.09	2.37 ± 0.22
Protein Efficiency Ratio	21	1.06 ± 0.14	1.09 ± 0.17	1.00 ± 0.24
	45	0.88 ± 0.04	0.90 ± 0.10	0.87 ± 0.08
	90	0.89 ± 0.07	0.88 ± 0.04	0.78 ± 0.06
VSI (%)	21	11.42 ± 0.96	12.38 ± 2.21	11.71 ± 1.97
	45	11.43 ± 1.35	11.52 ± 1.17	11.16 ± 1.32
	90	11.01 ± 1.21	11.00 ± 2.11	10.08 ± 1.42
HSI (%)	21	2.22 ± 0.43	2.39 ± 0.57	2.23 ± 0.39
	45	2.45 ± 0.77	1.92 ± 0.35	2.26 ± 0.48
	90	2.00 ± 0.37	2.01 ± 0.25	2.22 ± 0.34
Coefficient of Fatness (%)	21	6.23 ± 0.97	6.34 ± 1.06	6.53 ± 1.63
	45	6.42 ± 1.00	6.86 ± 1.06	6.38 ± 1.03
	90	6.48 ± 1.21	6.53 ± 1.95	5.47 ± 1.32
K	21	0.50 ± 0.20	0.47 ± 0.19	0.56 ± 0.23
	45	0.82 ± 0.41	0.59 ± 0.37	0.79 ± 0.45
	90	0.75 ± 0.51	0.74 ± 0.29	1.15 ± 0.65
Relative Intestinal Length	21	0.52 ± 0.06	0.56 ± 0.09	0.52 ± 0.07
	45	0.49 ± 0.13	0.52 ± 0.12	0.50 ± 0.07
	90	0.43 ± 0.13	0.42 ± 0.08	0.38 ± 0.10

**Table 4 animals-13-03383-t004:** Mean value ± standard deviation of the histological parameters of the proximal gut measured in sea bass fed the three experimental diets supplemented with *Saccharomyces cerevisiae* var. *boulardii* (LSB) at a concentration of 0, 100 or 300 mg/Kg of diet, respectively.

	Days of Feeding								
	21			45			90		
	LSB0	LSB100	LSB300	LSB0	LSB100	LSB300	LSB0	LSB100	LSB300
VL (µm)	381.54 ± 97.22 ^ab^	336.39 ± 90.23 ^b^	481.83 ± 137.26 ^a^	267.67 ± 70.96	279.69 ± 83.54	278.56 ± 91.11	299.54 ± 73.23 ^ab^	345.42 ± 72.26 ^a^	278.38 ± 37.28 ^b^
VT (µm)	86.65 ± 20.39	92.28 ± 19.85	102.12 ± 14.90	82.41 ± 14.40	95.01 ± 28.82	82.54 ± 17.68	84.73 ± 18.01	95.39 ± 22.24	86.54 ± 11.67
GC	5.07 ± 4.41	6.92 ± 3.05	4.33 ± 8.80	3.07 ± 2.50	3.23 ± 2.85	3.5 ± 2.59	2.85 ± 1.90	2.5 ± 2.01	2.5 ± 1.57
LP (µm)	8.68 ± 4.05	10.8 ± 1.28	8.77 ± 2.52	10.89 ± 1.78	11.32 ± 2.91	10.8 ± 2.07	10.78 ± 2.04	12.55 ± 3.46	12.12 ± 2.01
MT (µm)	3.48 ± 0.64	3.44 ± 0.52	4.11 ± 1.34	4.02 ± 0.64	3.92 ± 0.82	3.7 ± 0.46	3.85 ± 0.77 ^b^	4.49 ± 0.80 ^a^	3.57 ± 0.41 ^c^
SL thickness (µm)	42.10 ± 16.10	47.28 ± 20.16	62.94 ± 28.42	63.46 ± 19.89	52.11 ± 18.18	60.41 ± 23.53	49.27 ± 15.20	45.7 ± 17.30	37.93 ± 13.21
ML thickness (µm)	89.63 ± 32.32	80.59 ± 32.05	110.99 ± 34.10	115.11 ± 37.60	91.89 ± 40.28	117.97 ± 34.84	97.69 ± 28.92	86.06 ± 35.83	93.87 ± 26.52
SML thickness (µm)	23.41 ± 6.62	29.89 ± 7.97	28.66 ± 10.02	26.69 ± 13.34 ^b^	36.28 ± 8.01 ^a^	32.59 ± 9.19 ^ab^	27.51 ± 11.12	30.78 ± 14.46	27.81 ± 9.76

Villi Length (VL); Villi Thickness (VT); number of goblet cells (GC); thickness of Lamina Propria (LP); Microvilli Length (MT); Serous Layer (SL) thickness, Muscular Layer (ML) thickness, and SubMucosa Layer (SML) thickness. Different letters in the same row, at the same timepoint, indicate the presence of statistically significant differences.

**Table 5 animals-13-03383-t005:** Mean value ± standard deviation of the non-specific immune parameters measured in the sea bass fed the three experimental diets supplemented with *Saccharomyces cerevisiae* var. *boulardii* (LSB) at concentrations of 0, 100, or 300 mg/kg of diet, respectively.

Parameter	Days of Feeding	Diets		
		LSB0	LSB100	LSB300
Skin mucus lysozyme (U/mL)	0	1.397 ± 0.120		
	21	1.384 ± 0.084	1.397 ± 0.103	1.397 ± 0.103
	45	1.421 ± 0.055	1.570 ± 0.081	1.570 ± 0.081
	90	1.137 ± 0.194	1.236 ± 0.200	1.236 ± 0.200
Plasma lysozyme (U/mL)	0	2.287± 0.105		
	21	2.184 ± 0.103	2.287 ± 0.084	2.328 ± 0.075
	45	2.522 ± 0.152	2.596 ± 0.071	2.596 ± 0.089
	90	2.497 ± 0.164	2.534 ± 0.122	2.534 ± 0.084
Kidney lysozyme (U/mL)	0	2.027 ± 0.167		
	21	1.669 ± 0.138	1.875 ± 0.098	1.936 ± 0.121
	45	1.730 ± 0.187	1.710 ± 0.098	1.916 ± 0.187
	90	1.730 ± 0.137	1.710 ± 0.197	1.916 ± 0.098
Hemolytic activity (SH 50 Units)	0	1.407± 0.650		
	21	1.470 ± 0.384	1.806 ± 0.487	2.528 ± 0.826
	45	1.000 ± 0.990	1.319 ± 0.630	2.271 ± 1.530
	90	3.140 ± 1.140	3.578 ± 0.922	3.680 ± 0.623
Hemagglutinating titer	0	4 ± 0		
	21	8 ± 2	8 ± 2	128 ± 42
	45	8 ± 2	8 ± 2	8 ± 2
	90	256 ± 32	32 ± 12	8 ± 0
Respiratory Burst (nmol O_2_/10^6^ granulocytes)	0	0.022 ± 0.010		
	21	0.060 ± 0.020	0.027 ± 0.010	0.033 ± 0.030
	45	0.387 ± 0.080	0.355 ± 0.020	1.771 ± 0.080
	90	0.175 ± 0.040	0.291 ± 0.010	1.931 ± 0.500
Peroxidase activity (U/mL)	0	0.063 ± 0.010		
	21	0.053 ± 0.006	0.087 ± 0.050	0.062 ± 0.020
	45	0.138 ± 0.080	0.091 ± 0.040	0.072 ± 0.020
	90	0.097 ± 0.020	0.202 ± 0.080	0.247 ± 0.020

**Table 6 animals-13-03383-t006:** Antibacterial activity measured in the serum, kidneys, intestinal and skin mucus of the sea bass fed the three experimental diets supplemented with *Saccharomyces cerevisiae* var. *boulardii* (LSB) at concentrations of 0, 100, or 300 mg/kg of diet, respectively. The obtained results are expressed in terms of the percentage of the total number of examined samples that showed a positive reaction.

			Antibacterial Activity			
	Days of feeding		*E. coli*	*P. aeruginosa*	*S. aureus*	*S. epidermis*
Serum	0	LSB0	0	0	0	0
	21	LSB0	0	0	0	100
		LSB100	0	0	0	0
		LSB300	0	0	0	100
	45	LSB0	0	0	0	0
		LSB100	0	0	0	0
		LSB300	0	0	0	50
	90	LSB0	0	0	0	50
		LSB100	0	0	0	100
		LSB300	0	50	0	0
Kidney	0	LSB0	0	50	50	0
	21	LSB0	50	50	50	0
		LSB100	0	100	0	0
		LSB300	0	50	0	0
	45	LSB0	0	100	0	0
		LSB100	0	50	0	0
		LSB300	50	0	0	0
	90	LSB0	100	0	0	0
		LSB100	0	50	0	0
		LSB300	0	50	0	0
Intestinal Mucus	0	LSB0	25	100	0	25
	21	LSB0	-	-	-	-
		LSB100	-	-	-	-
		LSB300	-	-	-	-
	45	LSB0	0	0	50	50
		LSB100	0	50	0	0
		LSB300	100	50	0	0
	90	LSB0	0	100	0	100
		LSB100	0	100	0	100
		LSB300	0	0	0	0
Skin Mucus	0	LSB0	-	-	50	-
	21	LSB0	-	100	100	-
		LSB100	-	100	100	-
		LSB300	-	100	100	-
	45	LSB0	-	100	100	100
		LSB100	-	100	100	100
		LSB300	-	100	100	100
	90	LSB0	-	100	100	100
		LSB100	-	100	100	100
		LSB300	-	100	100	100

**Table 7 animals-13-03383-t007:** Mean value ± standard deviation of the gene expression of the pro-inflammatory cytokine *il*-1*β*, *il*-6, and *tnf-α* genes in the proximal gut of the sea bass fed the three experimental diets supplemented with *Saccharomyces cerevisiae* var. *boulardii* (LSB) at concentrations of 0, 100 or 300 mg/kg of diet, respectively.

	Days of Feeding								
	21			45			90		
	LSB0	LSB100	LSB300	LSB0	LSB100	LSB300	LSB0	LSB100	LSB300
*il-1β*	1.29 ± 0.24	1.33 ± 0.19	1.19 ± 0.43	1.02 ± 0.25 ^ab^	1.11 ± 0.3 ^a^	0.59 ± 0.33 ^b^	0.87 ± 0.28	0.73 ± 0.47	0.67 ± 0.18
*il-6*	0.67 ± 0.15 ^a^	0.46 ± 0.13 ^b^	0.26 ± 0.12 ^c^	0.44 ± 0.33	0.29 ± 0.11	0.31 ± 0.12	0.35 ± 0.09 ^c,b^	0.4 ± 0.22 ^b^	0.65 ± 0.13 ^a^
*tnf-*α	0.86 ± 0.49	0.91 ± 0.42	0.85 ± 0.61	0.75 ± 0.37	0.57 ± 0.38	0.41 ± 0.31	0.73 ± 0.43	0.87 ± 0.32	0.72 ± 0.62

Differences between the data series were determined by means of the Kruskal–Wallis test; different letters in the same row, at the same time point, indicate the presence of statistically significant differences.

**Table 8 animals-13-03383-t008:** Mean value ± standard deviation of the gene expression of the pro-inflammatory cytokine *il*-1*β*, *il*-6, and *tnf-α* genes in the distal gut of the sea bass fed the three experimental diets supplemented with *Saccharomyces cerevisiae* var. *boulardii* (LSB) at concentrations of 0, 100 or 300 mg/kg of diet, respectively.

	Days of Feeding								
	21			45			90		
	LSB0	LSB100	LSB300	LSB0	LSB100	LSB300	LSB0	LSB100	LSB300
*il-1β*	1.07 ± 0.06 ^a^	0.63 ± 0.21 ^b^	0.83 ± 0.45 ^ab^	0.99 ± 0.2	1.03 ± 0.18	0.68 ± 0.59	0.76 ± 0.08	0.95 ± 0.52	0.78 ± 0.13
*il-6*	0.79 ± 0.29	0.74 ± 0.17	0.83 ± 0.33	0.88 ± 0.32 ^ab^	1.02 ± 0.21 ^a^	0.70 ± 0.2 ^b^	0.76 ± 0.25	0.70 ± 0.06	1.04 ± 0.46
*tnf-*α	1.1 ± 0.42	1.25 ± 0.36	1.21 ± 0.33	1.16 ± 0.36	0.79 ± 0.38	0.89 ± 0.36	0.90 ± 0.33	1.18 ± 0.37	1.01 ± 0.5

Differences between the data series were determined by means of the Kruskal-Wallis test; different letters in the same row, at the same time point, indicate the presence of statistically significant differences.

## Data Availability

The data that support the findings of this study are available from the main author, A.P., upon reasonable request.

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
