# Peer review of "Live Yeast (Saccharomyces cerevisiae var. boulardii) Supplementation in a European Sea Bass (Dicentrarchus labrax) Diet: Effects on the Growth and Immune Response Parameters"

_animals, 2023, doi:10.3390/ani13213383_

Round 1

Reviewer 1 Report

Comments and Suggestions for Authors

Comments to author

Re: animals-2569369 “Live yeast (Sacchromyces cerevisiae var. boulardii”) supplementation in European sea bass (Dicentrarchus labrax) diet: Effects on growth and immune response

This manuscript presents original data resulting from a study on the effect of feeding a live yeast organism to the zootechnical performance and immune response parameters on sea bass. The data presented in the manuscript is novel and interesting for publication; however, the paper requires revision in areas outlined below:

The simple summary only outlines the effects of live yeast supplementation on gene expression responses, what about all other parameters measured here.

Line 33-39 there is a focus on the gene expression results and lack of summarising the effects of live yeast supplementation on all other parameters measured. Please include analysis here.

Line 51-54 the authors need to be more explicit on the use of probiotics in animal feeds, are they referring to the prophylactic use of probiotics to reduce antibiotic use, is this approach going to replace antibiotic use or complement in contemporary disease strategies.

Line 64-66 this statement doesn’t make sense considering the authors have in the previous paragraph referenced and described many advantages of probiotics from in vivo studies.

Lines 74-78 be more explicit “strong responses from these chemical messengers” what does this mean in the context of the immune response, are we talking innate immunity, adaptive immunity or both? 

In general, the introduction does require to be more explicit its not enough to justify the use of probiotics to modulate the immune response without further dissection of the information referring the reader to the effects of probiotics on innate or adaptive immunity. Length of exposure of probiotics and delivery of probiotics are important factors effecting the efficacy on the immune host responses. None of this was addressed in the introduction.

lines 121-122 concentration, duration of exposure?

Line 135 – no inclusion of Levucell in the table please add

Line 147 -152 why was the initial inclusion at 1010 then after analysing in diets dropped to 105? This is a big loss of yeast probiotic in the diet preparation!

Line 159 was it daily feeding rate if you only bulk weighed once a month?

Line 162-169 lack of detail in description of method was it sterile conditions i.e. aseptic? What was the concentration and pH of diluent used? Decimal dilutions this is unclear what was performed, was the faecal material weighed to ensure a 10-1 dilution was prepared correctly before performing the serial dilution to determine the CFU/g of faeces. What concentrations were plated etc. The current presentation is too vague and not reproducible.

Line 187-189 unclear why these pathogens were used for the antibacterial assays.

Line 190-191 what was the samples stored in for the RNA extractions.

Line 224 Bouin solution what was the pH and concentration of this solution.

Line 270-283 how much serum used was it filtered to remove artifacts such as skin debris, scales etc….how many replicated were used per treatment? What was the concentration as duration of exposure to stimulants Zymosan A and PMA as these are potent stimulators of macrophages from M1 to M2 phenotypes

Line 290 – what was the thickness of agar on plates did this conform to the Kirby-Bauer test conditions?

Line 291-292 how did the authors determine the concentration of each tested pathogen?

Line 310 – 320 these sections lack detail and require revisions, what QC was performed on RNA before the RT reaction. What was the average concentration of RNA used in the RT reactions? Was this normalised and to what concentration to remove sample extraction bias. What where the PCR conditions of the RT reaction. What primers were used in the RT reactions to generate the cDNA from RNA? Were all primers used in the experiment QC checked to ensure the efficiency of each primer was close to optimal efficiency as the delta delta CT method assumes a 100% efficiency and so primer bias is not accounted for. Why only used one reference gene? Is this experimental model a robust model can the authors provide evidence that the use of one refence gene is stable enough to ensure the stability of each reaction and remove technical bias.

Line 332-333 why use SD for some parameters and SE for others?

Comments on the Quality of English Language

Generally the quality of English is good

Author Response

Dear Reviewer,

We would like to thank you for the revisions of our paper ID: 2569369 Animals “Live yeast (Saccharomyces cerevisiae var. boulardii) supplementation in European sea bass (Dicentarchus labrax) diet: Effects on growth and immune response parameters” .

All of your comments were duly considered and all of them were accepted. Revisions were included in the manuscript and highlighted in red. Attached, we provide our point-by-point response.

Reviewer 2 Report

Comments and Suggestions for Authors

The presentation of data and the way of writing need so much effort and is far away from a standard scientific manuscript. Further, some data does not look alright, and they need to check all data again. Due to these issues, I did not go through line-by-line comments.

Comments on the Quality of English Language

Moderte improvment

Author Response

Dear Reviewer,

The authors thank you for the review to (paper ID: 2569369 Animals) Live yeast (Saccharomyces cerevisiae var. boulardii) supplementation in European sea bass (Dicentarchus labrax) diet: Effects on growth and immune response parameters”.

We would have appreciated to know comments and revisions of Reviewer 2 that would have surely helped us to improve quality of the paper. Anyway, we hope that after the changes made, the revised text has become more clear than in the previous version.

Reviewer 3 Report

Comments and Suggestions for Authors

Manuscript ID: animals-2569369

Title: Live yeast (Saccharomyces cerevisiae var. boulardii) supplementation in European sea bass (Dicentrarchus labrax) diet: Effects on growth and immune response parameters

The present manuscript is very well written and discussed, especially in the introduction and discussion sections. However, there are some doubts that I would like to be resolved with this review, mainly regarding the Material and Methods section. Below are my considerations:

Material and methods

Line 131: Please add the mean total fish length and its standard deviation.

Lines 148-149: How were these concentrations defined? Is there any previous study that tested these same concentrations? Please explain.

Line 157: Did the water temperature range from 15.6 to 26.6°C? Why such a large variation if the test took place in the laboratory?

Line 182: Please specify which mucus (skin and intestine?).

Lines 190-191: How did you differentiate between the proximal and distal portions during intestinal collection?

Line 217: What viscera were considered in this parameter? Please include in the manuscript.

Results

Table 4: Arrange the * in the table, placing them close to their respective means. Enter the standard deviation of the means and organize the table so that all values have 2 places after the decimal point.

Discussion

Lines 562-564: Mean water temperature varied by about 11°C in their study. This should be considered in the discussion, as immune parameters can be highly affected by changes in environmental cultivation conditions.

Author Response

(The authors gave the same response as above.)

Reviewer 4 Report

Comments and Suggestions for Authors

Simple Summary

This section is well prepared. Define ESB

Abstract

This section is well prepared.

Introduction

This section is well prepared.

Material and methods

Was this study performed 10 years-ago? If so, the validity of the study could be questioned as the answer may change in relation to the genetic selection of fish from that date to the present and 10 years preserved samples. Please clarify this point.

Line 120 “Photoperiod and thermoperiod were natural.” Include more detail regarding photoperiod. Were there thermal fluctuations during the experimental period? If so, how did these temperature variations affect the conduct of the experiment? Clarify this aspect.

Lines 120-121 “Water quality parameters were maintained in a range suitable for European sea bass” Include values (means and SD).

Line 198 “Kidney was weighted and 0.5 mg” Is this value correct?

Lines 232-233 “The measures and observations adopted in this study were based on a combination of the criteria previously given by different authors” Briefly include an explanation of these measurements.

Statistical analysis for gene expression must be done with Kruskal-Wallis test.

Results

3.5. Immune Innate Response. This subsection should be rewritten regarding significant differences, not only indicating maximum or minimum values.

Table 3 must include superscripts to highlight the statistical differences between the treatments.

Genes abbreviations must be written in lowercase and italics.

For the statistical analysis shown in Tables 7 and 8, I recommend analyzing with the Kruskal-Wallis test and a posthoc Nemenyi test and avoiding using the Mann-Whitney test used for two treatment comparisons. Correcting this aspect and rewriting the section according to the statistical differences is necessary.

Discussion

Although the authors reference the use of probiotics and their effects on different fish species, they include references to bacteria and their beneficial effects; however, the authors use yeast, so it is necessary to change this type of reference and use those where yeasts are added.

The authors show many comparisons with various fish species, which is correct, although, as I mentioned earlier, it is necessary to compare against studies performed with yeast supplementation and not with bacteria. In this aspect, I would like the authors to integrate their most relevant results and explain the immunological and metabolic mechanisms of using S. cerevisiae in diets for D. labrax and their importance during its culture.

Conclusion.

I recommend reviewing this section after attending the comments and suggestions.

Author Response

Dear Reviewer,

We would like to thank you for the revisions of our paper ID: 2569369 Animals “Live yeast (Saccharomyces cerevisiae var. boulardii) supplementation in European sea bass (Dicentarchus labrax) diet: Effects on growth and immune response parameters”.

All of your comments were duly considered and all of them were accepted. Revisions were included in the manuscript and highlighted in red. Attached, we provide our point-by-point response.

Round 2

Reviewer 2 Report

Comments and Suggestions for Authors

Unfortunately, they could not improve the quality of the MS and I suggest authors to read and read and then start to write an MS on this topic. Lack of knowledge has caused this MS to be far away from standard. I did not go through line be line comments as is so much issue with that.
Kind regards

Comments on the Quality of English Language

Revision is required.

Author Response

We would have appreciated to know comments and revisions of Reviewer 2 that would have surely helped us to improve quality of the paper. Anyway, we hope that after the changes made, the revised text has become more clear than in the previous version.

Reviewer 4 Report

Comments and Suggestions for Authors

It is a bit strange the way you are numbering the references. Normally, they are cited as they appear from the introduction and in ascending order, so you should not reference high numbers that later appear in the discussion. Check the numbering and make the necessary adjustments.

Author Response

Dear Reviewer,

thanking you for your time, we would like to ask additional explanations to your previous comment, because we did not understand the indications. Could you please give us more details?

Kind Regards,

Martina